# Creating Family-Centred Support for Children with Developmental Disabilities in Africa: Examples of Local Community Interventions

**DOI:** 10.3390/ijerph21070925

**Published:** 2024-07-16

**Authors:** Roy McConkey, Sally Allen, Chipo Mlambo, Patricia Kambarami, Karina Martin

**Affiliations:** 1Institute of Nursing and Health Research, Ulster University, Belfast BT15 1ED, UK; 2Templer Foundation, 6900 N. Haggerty Road, Canton, MI 48187, USA; sallyallen79@gmail.com (S.A.); chipo@templerfoundation.org (C.M.); 3St Christophers, Hatcliffe Community Hall, Harare, Zimbabwe; director@stchristopherschildren.com; 4Nzeve, Box 3396 Paulington, Mutare, Zimbabwe; execdirector@nzeve.org

**Keywords:** disability, preschool, children, family-centred, community-based, low income, inclusion, healthcare, education

## Abstract

Preschoolers with disabilities and their family caregivers are overlooked by many public health initiatives, especially in low-income countries. Yet they can benefit from early intervention to promote their development soon after birth and to provide a better quality of life for their families. In this paper, we describe how a community-based approach has been implemented with minimal funding in two areas in Zimbabwe: a township in Harare and in rural areas of Manicaland Province. Our aim in sharing this information—allied with references to research studies recently undertaken in Africa—is that it will enable similar support to be replicated in other communities by local personnel. A logic model is used to describe the situation in which the two projects work, the various inputs they have provided to their community, and the different forms of support they have offered to the children and their caregivers. The project outputs are listed in terms of the number of beneficiaries helped and the activities undertaken. The outcomes achieved for the children, families, and communities are reported. The sustainability and extensions of community-based projects to address unmet needs are discussed. The main conclusion is that disadvantaged communities can be energised to address the needs of their most marginalised residents.

## 1. Introduction

Children with disabilities and their families are among the most marginalised in African communities. The stigma associated with their impairments often leads to their exclusion from family and community life. They are denied access to health services that are often provided in distant clinics and hospitals. Equally, their chances of attending school are much reduced. Families have an even lower quality of life than their peers, often due to poverty and substandard housing compounded by the poorer physical and mental health of caregivers [1]. Indeed, this, too, is the reality for families who have a disabled child in other continents and even those living in low-resourced communities in high-income nations.

To date, their needs have been neglected in efforts to improve the global health of nations, perhaps because the strategies employed in promoting public health are inadequate to the challenges presented by these children and their families [2]. A favoured strategy in global health has depended on high-level policy formation and its implementation using a ‘top-down’ approach, an approach that has had mixed results for disabled persons [3]. Instead, a community-based approach, centred around grassroots responses to the needs of persons with disabilities, has been promoted [4]. The basic assumption is that local people have assets available to them to tackle the difficulties they face [5]. But how realistic is this approach in low-income countries in Africa and elsewhere?

In a companion paper, we undertook a rapid review of the published literature, which identified 25 review papers incorporating over 1500 empirical studies worldwide [6]. This suggested that there are low-cost, evidence-based strategies to address the needs of these children and their caregivers that can be implemented in communities by local people. Five key features were identified. First, the leadership functions required to create and implement the support services. Second, the provision of family-centred, home-based support to caregivers. Third, encouraging peer support for families. Fourth, mobilising the support of significant groups within and outside the community, and fifth, promoting children’s inclusion in early child development centres and primary schools.

In this paper, we describe how a community-based approach has been implemented with minimal funding in two areas in Zimbabwe: A township in Harare and in rural areas of Manicaland Province. Our aim is to share the experiences and activities that were undertaken through these community initiatives, which are rarely reported in the extant literature. Our aspiration is that this information—allied with references to research studies recently undertaken in Africa—will enable similar support to be replicated in other communities by local personnel. The authenticity of these accounts—their shortcomings as well as successes—offers hope that a better life is possible for those on the margins of their communities.

Ironically, our intended readership is unlikely to read articles in academic journals. Rather, copies of this and other articles we reference can be freely downloaded, and their content can be passed on to community personnel by the teaching staff in universities, colleges, and health and social services. It is also our hope that policymakers might be influenced to consider how they can incorporate community-based approaches into their public health strategies.

### 1.1. Country Context

Zimbabwe is a land-locked country in southern Africa with a population of 15.18 million in the 2022 National Census and a life expectancy of 64.7 years [7]. The Gross National Income in 2022 was US$1710. In all, 61.4% live in rural areas. Shona was spoken by 80.9% of the population in their early childhood, and for 11.5%, it was Ndebele. The employment to population ratio was 27.7% at the national level: highest in the capital Harare at 41.6%, but lower in rural provinces. The Under 5 mortality rate was 39.7 per 1000 live births.

The Washington Group Six Questions on functioning difficulties were used in the 2022 Census to identify children who could be considered to have a disability. (Details available at: https://www.washingtongroup-disability.com/question-sets/wg-short-set-on-functioning-wg-ss/1, accessed on 1 July 2024). Those rated as having ‘a lot of difficulty’ and ‘cannot do at all’ on any of the six functions—seeing, hearing, walking, remembering, self-care and communication—were classified as having a disability. In all, 0.5% of the 5- to 17-year-old population met these criteria, suggesting that over 35,000 children nationally could be considered disabled. However, when the criteria were widened in an earlier national household survey to cover children with milder forms of disabilities, the prevalence rate rose to 10% [8]. Potentially, there could be upwards of 1 million Zimbabwean children with developmental delays and disabilities.

### 1.2. Community Context

Two autonomous support services for families and children with disabilities are described in this review. Both are registered with the Government as Private Voluntary Organisations (PVOs) and are also known as Non-Governmental Organisations (NGOs).

St Christophers (details available at: https://stchristopherschildren.com/, accessed on 1 July 2024) works in Hatcliffe, a poor and marginalized suburb on the northern outskirts of the capital Harare. The only official census statistics are available for Harare as a whole, but the Hatcliffe community is likely to be among the more impoverished, with high levels of unemployment, poor housing, lack of water and electricity, a high Under 5 mortality rate, and lower life expectancy for its residents.

Nzeve (details available at: https://nzeve.org.zw/, accessed on 1 July 2024) is based in the town of Mutare but aims to cover the Eastern Province of Manicaland. In 2022, the total population of the Province was 2.04 million persons with an average household size of 4.1. Around 80% lived in rural areas, with 57% of households in traditional dwelling units; 78% had no water in their premises, and 50% had no access to electricity. In all, only 22% of the adult population were in employment with the majority working in agricultural occupations. The Under 5 mortality rate was 50.8 per 1000 live births, and 0.58% were classified as having a disability.

## 2. Early Intervention through Family-Centred Supports

The two support services operate independently and have different origins.

Nzeve started 20 years ago with a focus on the well-being and improvement of the deaf community. An increasing number of parents whose children had other developmental disabilities came looking for assistance, so in 2019, they widened their early intervention services to work with parents and help them to know their child’s disability, how to assist and support their child so that they can live their lives as independently as possible. To date, 237 children with sensory impairments have been assisted, and 14 with developmental disabilities are currently supported in 2023.

St Christophers was started by a mother of a child with a disability who had a background in social policy (the fourth author: PK). Her own lived experience confirmed the gaps in both the private and public healthcare systems. Meeting strong and fearsome women on her journey convinced her of the need to support lonely families carrying multiple burdens. St Christophers has been operating for five years in Hatcliffe, specifically to assist families who have a child with developmental disabilities. Their purpose is to ensure that each child develops to their full potential by empowering caregivers and their families while advocating and implementing within their limited resources the human rights of children with disabilities, as enshrined in the Zimbabwean Constitution (2013). A total of 136 children were assisted in 2023.

Despite their separation geographically and organizationally, they have evolved very similar strategies in responding to the needs of families, and Figure 1 illustrates them in the form of a logic model. The situation column summarises the points made in the introduction above and expands on in the rapid review paper [6]. The remainder of the current article will provide further information on each component in the logic model.

## 3. Service Inputs

The listings under the input column broadly reflect the steps each service followed, with those on the left side describing the foundations of their work and on the right-hand side how the different supports emerged.

### 3.1. A Simple Guide to Early Intervention

In 2016, Sally Allen—a Zimbabwean resident and educator—published an accessible guide for use by parents of children with intellectual challenges based on over 30 years of experience of work on early intervention for families in Zimbabwe and internationally [9] and on evidence from the global literature [10]. The Guide is a primer on the essential elements of promoting children’s development from the earliest months. Although it could be used by parents on their own, the main purpose is to provide a family-support worker with resources they can work through with parents or caregivers, especially those who have little or no education and limited literacy skills.

The content is arranged into four Units: The first describes stimulating the child’s activities through play, the second encourages mobility, the third describes communication and talking, and the fourth describes personal care. The introduction provides guidance on interacting with the child.

The same pattern is followed in each unit. The steps that children go through in learning the skills within that area are described. Parents can evaluate their child’s progress along the development path and choose the next step to take. Ideas are listed for families to try to help their child to progress. Figure 2 shows an example of self-feeding.

The Guide is written in simple English with colourful illustrations. It has been translated into Shona, Ndebele, and Spanish. It was first field-tested in another disadvantaged suburb of Harare, where 10 peer educators were trained in its use with 50 families. (A short video is available on YouTube: https://www.youtube.com/watch?v=Oy-Q9ll65lc, accessed on 1 July 2024) The Spanish version is also used in Nicaragua within a family-support programme involving volunteer facilitators and over 100 families.

The Guide was introduced to St Christophers by the third author (CM), who was a former colleague of Sally Allen and had been involved in its field testing in Harare. She, in turn, was invited by Nzeve to train their staff in the use of the Guide within their service. Thus, an indigenously produced, accessible guide was a valuable tool in encouraging the existing community-based organisations to incorporate or extend their family-centred support within their services. Similar guides, manuals, and training materials have been produced internationally and adapted for use in other African countries, for example in Ethiopia [10,11].

### 3.2. Leadership of Local People

The personnel leading the support services had lived and worked in their communities and had personal experience of the challenges facing the residents, particularly the lack of integrated support to meet their needs. The leaders drew on their academic knowledge, past research, and professional experience in the field of children with disabilities to develop a range of support to meet their needs, as described below. Equally, they had identified a gap in their services which led them to willingly incorporate the home-based, early intervention service in the case of St Christophers when approached or to seek out support to commence a similar service in the case of Nzeve.

In both organisations, the directors recruited a part-time project manager to manage and coordinate family-centred support. Their role included the recruitment and training of two paid home visitors in St Christophers and, in the case of Nzeve, volunteer mothers of children with disabilities willing to mentor other families. The project managers were also involved in starting other activities or ensuring the families in early intervention had access to the other community services on offer. It is noteworthy that in each service, the managers came from a community development or educational background rather than from health services.

### 3.3. Funding from Donors

No fees are charged to the families by either organisation for the support they provide, bar a small contribution to the costs of any prosthetics and orthotic devices provided in order to encourage better care of them by families.

Instead, St Christophers’ early intervention is financially supported by the Templer Foundation based in the US, which had been started by a Zimbabwean in memory of his daughter with disabilities. The Templer Foundation recently extended its funding to cover medical expenses (see below). Their total expenditure in 2023 was around US$8000. This covered the salaries paid to the two home visitors who worked part-time on a pay scale equivalent to a public school teacher along with the expenses described below. The Director and Project Manager receive a small honorarium and fuel expenses. Other aspects of the community services provided by St Christophers are funded by grants from the Oak Foundation and those obtained online through their website. Most of these funds go towards direct service provision to children and families rather than salaries or administration.

Nzeve had received funding for their family-support service from an overseas donor up to 2022, but they managed to maintain a reduced service through 2023 and have since secured funding from 2024 onwards from a German charity, Bread for the World, augmented by other donations garnered through their website. Their estimated expenditure on the family support in 2023 was around US$20,000.

In both services, the number of families who can be offered the home-visiting element of the service is determined by the available income. However, donors appear more willing to contribute to an established service, as their applications to various donors for start-up costs are often refused.

### 3.4. Home Visitors

The two home visitors recruited by St Christophers live in Hatcliffe; one was a former community health worker on an AIDS project, and the other had a degree in community development. Nzeve home visitors are mainly mothers of children with disabilities who volunteer to mentor and support other families under the guidance of the project manager.

The third author (CM) acts mainly as a volunteer project manager at St. Christophers. She trained their home visitors and also the Nzeve staff in the use of the Simple Guide described above. The training also covered the ethos underpinning a family-centred service and the importance of building trusted, mutually respectful relationships with families.

In Hatcliffe, on average, 16 families receive a home visit each month, with the home visitors travelling on foot or by public minibus. They also have further contact with families during clinic sessions or via WhatsApp calls and messages. The project manager supervises their work by accompanying them on occasional home visits through WhatsApp reports and attending clinic sessions.

At Nzeve, home visits are conducted when clinic staff identify a need. For example, when the caregiver and the child have missed multiple sessions, or a staff member is worried about the progress of the child. In most cases, parent mentors will travel by bus or the organisation’s vehicle to conduct the visit. At times, the parent mentor will travel with Nzeve staff. Nonetheless, Nzeve found that parent mentors are the best people for conducting such visits, as they have lived experience that allows them to identify more with the caregiver of the child. After the visit, a report is written, and a follow-up is made to ensure that the issues have been resolved.

### 3.5. Partnerships with Health and Education

The barriers encountered by people with disabilities in accessing primary healthcare services in low-income countries have been well documented in the literature, notably cultural beliefs or attitudinal barriers, informational barriers, and practical or logistical barriers [12]. Hence, both organisations have been active in overcoming the barriers by advocating for children and families with government health and education departments. St Christophers approached the Ministry of Health and Child Care, requesting them to provide an outreach clinic one morning a week by a minimum of six therapists from the rehabilitation team at the Referral Hospital.

Contact was also made with the medical school at the University of Zimbabwe with a view to having a trainee paediatrician undertake a similar clinic in Hatcliffe. This request was granted, and families known to St Christophers are provided with medical examination, medications, or referrals to other specialist services.

International research has confirmed the heightened risks of undernutrition in early childhood among very young children with disabilities in low-income countries [13], and this was evident in families in Harare. The Malnutrition Unit of Sally Mugabe Children’s Hospital was invited to screen and assess the nutritional status of children with disabilities in Hatcliffe once every two months. In total, 32 were attended to in 2023. The unit also provided education, training, and demonstrations on good nutrition, correct feeding positions for children with cerebral palsy, and nutritional counselling.

The project manager and St Christophers also requested the Schools Psychological Services, under the Ministry of Primary and Secondary Education, to assess children who were out of school and those who were approaching the age to start primary school. They agreed and even undertook their assessments in the community hall in Hatcliffe, a familiar venue for children and families.

Nzeve is progressing with a memorandum of understanding (MOU) with the Ministry of Health and Child Care, which will allow them to work with various hospitals around the country to conduct more early identification outreaches. Additional funding from the Oak Foundation has enabled them to perform hearing test screening in a new community service in Chipinge town.

### 3.6. Family-Centred Support

In Nzeve, the children are first identified at the developmental clinics that undertake hearing assessments. Those aged under 7 years who have other disabilities—mostly Cerebral Palsy and Down Syndrome—can still avail of family-centred support, which is mostly provided at their clinic, with transport provided to families.

St Christophers’ home-visiting service aims to support children with significant and any impairments, mostly up to six years of age. Here, too, most of the children have cerebral palsy. Families hear about St Christophers by word of mouth from other families or community workers. They are encouraged to meet the St Christophers team for therapy sessions and possible home visits if the child is under 6 years of age. If the caregiver wishes to join, further home visits are undertaken.

Home visits last between 60 and 90 min and usually take place monthly at first but reduce over time. Initial visits are used to learn more about family circumstances and to assess the child’s development. Together, home visitors and caregivers work through the Simple Guide, identifying appropriate goals and activities for families to use at home. The home visitor can demonstrate suitable activities using whatever objects and equipment are available. The overarching goal is to have family members interact with the child in a responsive and enjoyable manner.

They can also provide a listening ear to caregivers as they talk about their personal worries and concerns. Caregivers can be signposted to organisations such as local social welfare community workers, which could assist with issues of mental health which neighbouring countries have reported as an unmet need [14].

A record is made of the goals chosen to advance the child’s development and the activities to be used to achieve them. These are noted in the family’s copy of the Guide and also recorded on WhatsApp messages sent to the project manager. On subsequent visits, the child’s progress is reviewed with the family when new or revised goals will be set. However, due to the limited literacy of caregivers, oral records are mainly preferred rather than formal written reports.

Home visitors and families can contact each other via WhatsApp messages or calls. They may also meet at clinic sessions or at parent meetings.

A similar process is followed by Nzeve, which is also centred around the Guide with identified goals recorded in the children’s clinic files.

### 3.7. Assistive Devices and Equipment

Both projects endeavour to provide assistive devices that they feel would assist the child. Through local and international partners, St Christophers facilitates the provision of assistive mobility devices, although presently, they have a large unmet need for some types of products for children. Likewise, Nzeve sells second-hand hearing aids and loans other assistive devices such as standing frames, corner seats, parallel bars, and push carts as required for families to use at home. After achieving the child’s developmental goal, the assistive devices are returned to Nzeve.

### 3.8. Access to Health Services

Following on from the agreement with the Ministry, St Christophers arranged with the rehabilitation team from the Referral Hospital to hold a weekly clinic in Hatcliffe attended by a physiotherapist, occupational therapist, speech and language therapist, and a rehabilitation technician. A room in the community hall is used, but mats, curtains, and equipment must be transported there each time. The therapists work in one room with the children and caregivers, who then move around the different therapies as needed. Home visitors arrange the rota for children and families attending the clinic and provide transport for those living in more distant places. Individual paper files are maintained to record the goals set for the children and the activities/exercises recommended by the therapists. These can be added to the individual plans for the child and family. Caregivers and children wait in an adjoining store-like room or on the veranda. This setting is not ideal, and a search continues for the project to have its own base in the community.

A similar arrangement works when the children attend a clinic held by a trainee paediatrician from the University of Zimbabwe. A fund has been established to cover the costs of medicines which are supplied through an account held by a local pharmacy. St Christophers also covers the costs of any specialist tests, such as EEGs.

Caregivers are informed and encouraged to take up government initiatives, such as polio and cholera vaccinations, mental health support, and other medical services, as a means of ensuring that children with disabilities are included in these programmes.

Nzeve has undertaken similar initiatives with local health services and has recently received funding that will enable them to cover the costs of a physiotherapist and occupational therapist on a consultancy basis.

### 3.9. Nutrition

St Christophers has undertaken various means to counter the malnutrition experienced by children and families. For example, a nutritious lunch is provided for children and caregivers attending the clinics, which is donated and cooked by a local chef. In addition, food hampers made up of donated food are distributed to needy families identified by home visitors.

Likewise, Nzeve provides lunches to both caregivers and children when they attend meetings. The produce is often from Nzeve’s own vegetable garden. Furthermore, the caregivers are mentored in nutrition training. Also, Nzeve has funded nutritional gardens for communities where people with disabilities live.

### 3.10. Parent-to-Parent Support

Both projects aim to promote mutual support among their caregivers. Nzeve has recruited and trained mothers to act as parent mentors who show support to other new parents through home visits. Additionally, they have become active advocates for the early identification of developmental disabilities and their inclusion in their respective communities. Training workshops are organised not only for parents but also for other family members. Among the topics covered have been nutrition, gender-based violence (GBV), safeguarding, and talks by health personnel on child development. In addition, a parent support group has been set up as a means of providing emotional support and information sharing. They use WhatsApp to ask questions and to share stories of support. A father’s support group has also met.

St Christophers has also provided training workshops for parents on pertinent topics, with invited speakers such as therapists, paediatricians, nutritionists, counsellors, education psychologists, and local leaders. The clinic sessions also provide further opportunities for families to meet one another. A social outing to a game park for over 50 parents and children also helped them bond as an example of recreational therapy. A WhatsApp group has been established and is well-used by caregivers.

The caregivers have created a group of their own and spoke of their intention to start self-sustaining projects, in particular, a daycare centre of their own. Their hope is to take turns caring for their children during the day, enabling mothers to go to work knowing their children are in the hands of someone they know and can trust.

### 3.11. Income Support and Generation

The alleviation of extreme poverty experienced by some families has inevitably had to feature in their work [15].

St Christophers has started to explore options for families to generate income. This includes starting a community garden with eight families and exploring options such as detergent manufacturing, peanut production, chicken and rabbit rearing, and selling second-hand clothes.

Nzeve has a longer history of providing vocational training programmes teaching horticulture, sewing, building, and carpentry, albeit with deaf youth in the main. Business training is also offered to all caregivers, both those with deaf children and those whose children have developmental delays. In addition, Nzeve has a separate project focusing on Income Savings and Lending Groups (ISALs) that specifically target persons with disabilities and their caregivers in rural communities [16]. These beneficiaries include other families identified from the community.

### 3.12. School Placement

Following the first set of assessments made by educational psychologists in Hatcliffe, three children were assessed and accepted for schools in other locations in Harare with fees, uniforms, and transport covered by St Christophers, thereby overcoming some of the barriers poorer families experience with respect to school attendance [17]. More recently, further assessments have led to a proposed plan to enrol 14 children in a special unit within a government school in Hatcliffe. St Christophers has started to raise funds for building a classroom and associated costs, as these are unlikely to be available from the Ministry of Primary and Secondary Education.

Nzeve has a longer history of working closely with the Ministry of Primary and Secondary Education, particularly the 18 schools across the Province that have Resource Units primarily for deaf learners. Children with developmental delays join mainstream schools once they have reached their developmental milestones. Nzeve supports schools and Resource Units through learning resources and disability inclusion training for teachers.

## 4. Outputs and Outcomes

Figure 1 summarises the outputs summated across the two projects in terms of the number of beneficiaries and/or the number of activities delivered within a 12-month period. These provide an indication of the diversity of support provided across the two communities at a relatively low cost of around US$28,000 per annum.

However, both projects rely more on qualitative indicators when it comes to describing the outcomes and impact of the support they have provided. In particular, their stories describe the changes in children and families. For example:


*When Perseverance started, she was 2 years 5 months. She was not able to sit alone. She had no head control and we had to feed her. Now she can feed herself and can walk alone. I am so grateful for these achievements that have happened, Perseverance is now 4 years 8 months.*



*St Christophers is my support family. When Riley first came to St Christophers, he had no head control, and he was very weak. I have been coming here for a year now and Riley has improved on head control, turns on his own and he has even gained weight. Thanks to the therapy, home visits, counselling and nutrition education that we receive at St Christophers. Even my relatives cannot believe the change that has happened to Riley.*



*Pearl has Down Syndrome; the daughter of deaf parents. When she first came to Nzeve she was unable to stand up unaided or to walk. She and her mother attended weekly communication sessions and inspired by the parent mentors, her father crafted parallel bars at home and within six months she was walking independently and playing with other children. Pearl’s success is a beacon of hope for the future.*


Project staff also reported similar improvements in the children’s motor, social, and communication skills. Other home-based interventions in Malawi [18], Zambia [19], and Ethiopia [20] have been able to document similar impacts on children’s development.

Previous research has recounted the stigma associated with disability in Zimbabwean society, especially accusations of witchcraft [21], but project staff noticed an impact their services have had on their communities.


*When people in the community see how children of their friends and families are improving, people change their perceptions and misconceptions about disabilities. People no longer see disability as associated with witchcraft and evil spirits and rather get to understand what disability really is.*



*The community councilors are more supportive of our project—they come to thank us for what we are doing.*


The projects also had disappointments. A few families have dropped out from home visits and clinic sessions. The main reasons have been the family moving out of the area due to the high price of rentals, caregivers being overwhelmed with the challenges they face, and others an apparent lack of motivation to help the child. Yet financial burdens and poor family support make it difficult for some parents to commit fully to therapy sessions and home visits.

Recruiting suitable staff is a continuing problem. St Christophers had difficulty in finding a suitable project manager. A promising appointment left after two months, and although seemingly committed to working with communities, she seemed overwhelmed. Finding suitable premises for the clinic sessions remains an unresolved issue. An overseas donor terminated their link with Nzeve. The ongoing need to raise funds is time-consuming. A constant frustration is not having the resources to extend support to more children and families.

## 5. Discussion

Three main conclusions can be drawn from the experiences of the two projects described here. First, it was possible to create family-based support services for children with disabilities in very disadvantaged communities in a low-income country such as Zimbabwe. In addition, examples from other countries have been cited from the published literature, which strengthens the assertion that these children and families can be helped and should no longer be ignored.

Second, the projects adopted remarkably similar strategies for supporting families despite being geographically separate and with differing origins. They employed many of the strategies confirmed in the rapid literature review [6], allied by their deep knowledge of the communities they had chosen to assist and their personal expertise and experiences of helping children with disabilities. The lesson for others is that communities can be trusted to make strategic decisions best suited to their needs.

Third, the projects rose to the challenge of providing more holistic support to children and families rather than just focusing on disabilities. Their ‘can-do’ attitude was greatly assisted by mobilising help from Government Ministries and other agencies through their persistent advocacy and negotiation. This was further evidence that people of goodwill are present throughout every human society and are predisposed to say ‘yes’ when asked for help.

However, the story of these community-based projects is not yet complete. The projects—and others like them—are fragile in two main respects. First, can they sustain the support they provide, and second, can they extend the support on offer? Hopeful responses can be given to both questions because the two projects have existed for five or more years, and for the immediate future, they seem financially secure. They also have a track record of expanding the range of support provided to families. Their successes build up the motivation and confidence of the leaders in particular, which, in turn, enthuses the families and communities. Hence, the continuity and extension of community projects depend on the nurturing of new leaders, most likely from existing personnel and beneficiaries [22]. Further training and career development opportunities for family-support workers will likely be needed [23].

The projects are also conscious of the needs that they have as yet been unable to meet, and which international experience suggests are vital for supporting families and children more effectively. This includes children’s access to early childhood care and education centres, either through their inclusion in existing facilities or by promoting the development of new centres. The benefits such centres bring to the child are well attested in terms of skills acquisition and peer relationships while increasing their readiness for attendance at primary schools [24]. Caregivers also gain through having time for income generation at home or through employment. Experience in neighbouring countries suggests that such centres need to be culturally relevant, with additional support given to help with the management of the learning needs of the preschoolers [25]. The Simple Guide with visiting support from home visitors are two assets the projects could offer to early childhood centres. Training courses could also be offered for centre staff.

Poverty alleviation is a further identified need that requires more sustained attention, as the evidence demonstrates the difference this makes to a family’s quality of life. In particular, cash transfers under governmental social protection schemes have been shown to be effective [16]. Partnerships with existing schemes are worthy of investigation, as they are allied with instigating income-generating projects for families and ensuring that safeguards are in place to prevent misuse of funds.

Strengthening the advocacy by caregivers and families is likely to be a necessary pre-requisite for gaining governmental support locally and nationally for the above endeavours. Certainly, its effectiveness in relation to improved access to healthcare, for example, has been demonstrated. Again, culturally sensitive strategies need to be devised to empower parents [26,27], with inspiration coming from stories of the impact families have had on ensuring that government policies are implemented [28].

Arguably, the most challenging question in relation to public health is this: can national coverage be attained through an aggregation of community-based initiatives? Given the sparsity of the latter in many low-income countries, it may be a premature question to ask, but nonetheless, one that community-based projects need to have in mind, especially with respect to helping other communities replicate the support they provide. Indeed, the two Zimbabwean projects are examples of sharing their training resources, and this article, and others cited in it, enables the learning they have accumulated to be passed on to many other practitioners in Africa and beyond. As more community-based projects develop, they could form networks among themselves to exchange information, knowledge, and stories of outcomes. The network might create training resources for use with paid and volunteer family supporters or to promote the empowerment of family members. As a network, they will become more powerful advocates with national governments in particular, around policy implementation and funding [29]. The following is a concluding thought. Ultimately, the delivery of government policy is dependent on its uptake by communities and their citizens. It is, therefore, of interest to nurture community-led actions if policy implementation is to be assured.

## 6. Conclusions

The challenges in responding to the needs of children with disabilities and their family caregivers can seem formidable. Yet, these two projects demonstrate that with commitment and enthusiasm from leaders of community-based, Non-Governmental Organisations and the help they sought from partners allied with the financial support from donors, they were able to improve the life chances of the children and families at a relatively low cost. Their example can serve as a model for other communities and, as such, is an important contribution to assisting local people to become active agents for community developments that address the needs of all their members.

## Figures and Tables

**Figure 1 ijerph-21-00925-f001:**
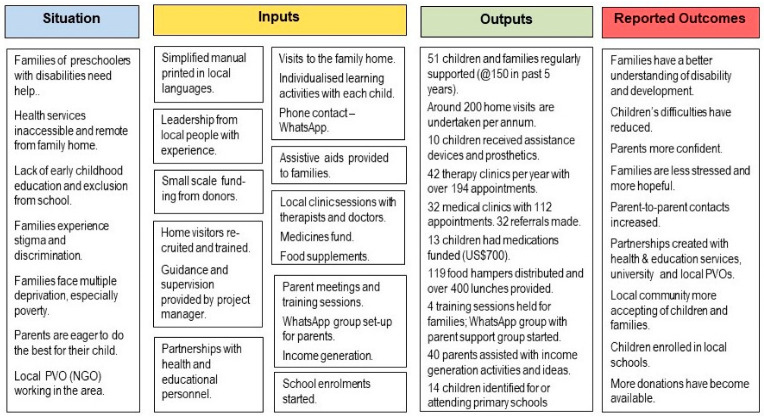
Logic model for the two family-centred support services.

**Figure 2 ijerph-21-00925-f002:**
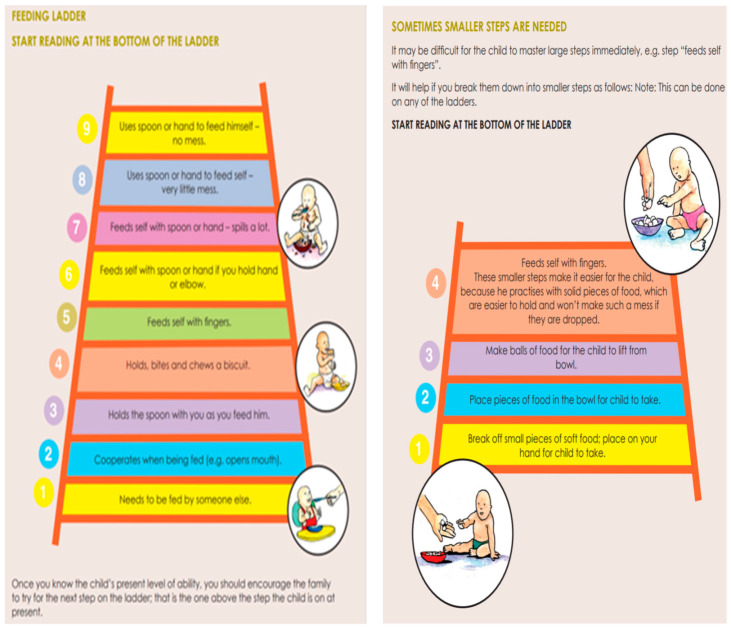
Example pages from the Guide.

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
