# Peer review of "Creating Family-Centred Support for Children with Developmental Disabilities in Africa: Examples of Local Community Interventions"

_ijerph, 2024, doi:10.3390/ijerph21070925_

Round 1
Reviewer 1 Report
Comments and Suggestions for Authors
Thank you for your work in this field and for the clarity of this paper. The overall paper is well written and easy to follow. Nicely done!
Very impressive that the community is doing this and not the govenrment. Unfortunaley once goverent gets involved, costs go up and the quality of servies changes.
The only edits I suggest are as follows:
Page 3 Line 118 - authors state that the Nzene work began 20 years ago and has served 237 children. Then you go on to say the St Christophers project served 136 children in just one year: 2023. How many children were served from the Nzene project in 2023? The overall number is valuable, but since you are comparing these two, tell the number from 2023 for both organizations.
Throuoghou the paper, there are spacing issues between sentences. Some appear to have up to 4 spaces between sentneces other just two. Make this consistent overall.
Page 4; Figure 1; Column 1: Under the first entry for 'Situation' there is listed "Rising prevalence of preschoolers with disabilties." I do not see support for this. Is it that there rising numbers or is it that there is more awareness of disabilities or disability catagories? I think perhaps 'rising awarenss' might be more accurate, or simply 'Preschooolers with disabilities'
Page 5; Figure 2 is very diffictul to read the tiny print. I worry if that is not fixed once this articile is published the very valuable tool will be rendered not so helpful.
Page 6; LIne 218: the word family should NOT begin with a capital letter F.
Page 2; Lines 70-75: Great point that those who need these services will not read the article, but the intent is for university professors to share this with their students! That acknowledgement is very critical to spreading the word!
Excellent contribution to the literature!
Author Response
Comment 1: Page 3 Line 118 - authors state that the Nzene work began 20 years ago and has served 237 children. Then you go on to say the St Christophers project served 136 children in just one year: 2023. How many children were served from the Nzene project in 2023? The overall number is valuable, but since you are comparing these two, tell the number from 2023 for both organizations.
Response 1. This has been clarified on lines 118/119
Comments 2: Throuoghout the paper, there are spacing issues between sentences. Some appear to have up to 4 spaces between sentences other just two. Make this consistent overall.
Response 2: This results from the justification of the text but could the copy-editor please check.
Comment 3: Page 4; Figure 1; Column 1: Under the first entry for 'Situation' there is listed "Rising prevalence of preschoolers with disabilties." I do not see support for this. Is it that there rising numbers or is it that there is more awareness of disabilities or disability catagories? I think perhaps 'rising awarenss' might be more accurate, or simply 'Preschooolers with disabilities'
Response: This has been changed to: "Families of preschoolers with disabilities need help."
Comment 4: Page 5; Figure 2 is very diffictul to read the tiny print. I worry if that is not fixed once this articile is published the very valuable tool will be rendered not so helpful.
Response: Copy editor please advise and address. I can supply the original pdf file from which the illustrations were taken.
Comment 5: Page 6; LIne 218: the word family should NOT begin with a capital letter F.
Response: This has been corrected
Comment 6: Page 2; Lines 70-75: Great point that those who need these services will not read the article, but the intent is for university professors to share this with their students! That acknowledgement is very critical to spreading the word!
Response: Thank you for the affirmation.
Reviewer 2 Report
Comments and Suggestions for Authors
Reviewer feedback
Abstract
Line 13 – are missed out in many public health initiatives – doesn’t make sense – over looked? Excluded from
Lines 23-30 4 sentences all begin with the word the
Introduction
Paragraph 1 – needs additional citations
Line 52-60 - the information from the literature review is very helpful – a table would be a beneficial graphic
Line 70 – this seems like an assumption or a biased position to take regarding the likeliehood of individuals to access academic journals
Iline 170 and 177 – both paragraphs start with the Guide. Is the guide evidence based on best practices for early intervention?
Line 196 – spelling error – family centered not centred
Line 251 – The inclusion of documented barriers strengthens to paper
Line 263 – This feels like an abrupt introduction of malnutrition? Was a needs assessment done in these communities with caregivers, Ngo representatives, and providers on the ground?
Line 306 = this model is reflective of early intervention services provided – the oral conversations are similar to individualized family service plans and should be referenced as such
Line 313- Were attempts made to connect with mission organizations that transport and manufacture medical equipment for more rural and underserved part of the world?
Line 354 – This manuscript would have been strengthened by formally collecting the qualitative data from parents on need for support and topics in which they felt support was needed? Again there are random mentions of gender based violence and a fathers support but no reference to prevalence of violence or number of fathers caregiving or participating in this group
Line 363 – the many references to Whatsapp are redundant – it would be better to include a paragraph addressing communication strategies
Lines 376-377 – I don’t understand how this relates to the larger article? Early intervention has been the focus but at this point income generation and vocational training is being discussed again without statistics addressing the need or linking it to early intervention
Lines 390 – What was the transition plan or process>?
Outputs and Outcomes – no methodology is disclosed for how these quotes were collected
The discussion lacks a comparison to any other currently established programs in these communities or other similar areas of the world – the goals of the initiatives are broad
Conclusion – I do not think the procedures or steps taken to establish these initiatives were clearly stated therefore limiting their impact as a model for other parts of the world
Comments on the Quality of English LanguageThe overall writing needs work on grammar, organization, and redundancy.
Author Response
Comment 1: Line 13 – are missed out in many public health initiatives – doesn’t make sense – over looked? Excluded from
Response: This has been changed to 'overlooked'.
Comment 2: Lines 23-30 4 sentences all begin with the word the
Response 2: This usage acts in lieu of bullet points to briefly summarise key messages.
Comment 3 Paragraph 1 – needs additional citations.
Response 3: No examples are given by the reviewer as to what citations are needed. However this is not intended to be an academic paper and the first paragraph is setting the scene. In any case we cite 29 pertinent articles throughout the paper and we reference a companion review paper (Citation 6) that published already by the Journal. .
Comment 4: Line 52-60 - the information from the literature review is very helpful – a table would be a beneficial graphic
Response 4: We have cited the paper which contains full details on each of the summary point made. We do not think a table is needed.
Comment 5: Line 70 – this seems like an assumption or a biased position to take regarding the likeliehood of individuals to access academic journals.
Response 5: From our 50 plus years of experience in both the developed and developing nations, this is more a statement of fact. We would be intrigued to know the reviewer's evidence for his/her comment.
Comment 6: Iline 170 and 177 – both paragraphs start with the Guide. Is the guide evidence based on best practices for early intervention?
Response 6: Yes and a note to this effect has been added on line 153. Apologies for this omission.
Comment 7: Line 196 – spelling error – family centered not centred
Response 7: This has been corrected.
Comment 8: Line 251 – The inclusion of documented barriers strengthens to paper
Response 8: Thanks for this affirmation.
Comment 9: Line 263 – This feels like an abrupt introduction of malnutrition? Was a needs assessment done in these communities with caregivers, Ngo representatives, and providers on the ground?
Response 9; The malnutrition was evident to the staff involved involved with the project so they wasted no time in trying to remedy it. This step has been noted on line 266.
Comment 10: Line 306 = this model is reflective of early intervention services provided – the oral conversations are similar to individualized family service plans and should be referenced as such.
Response 10: They are similar but also very different in how individual family plans are designed and recorded in affluent countries, so a citation to them would be misleading.
Comment 11: Line 313- Were attempts made to connect with mission organizations that transport and manufacture medical equipment for more rural and underserved part of the world?
Response 11: Yes and it was noted in the description of the projects that both have support from international agencies and their contacts.
Comment 12: Line 354 – This manuscript would have been strengthened by formally collecting the qualitative data from parents on need for support and topics in which they felt support was needed? Again there are random mentions of gender based violence and a fathers support but no reference to prevalence of violence or number of fathers caregiving or participating in this group
Response 12: We would have loved to do what the reviewer suggests but the resources to undertake such a studies were not available or worse still, had we diverted project funds to do so, it would have meant reducing the support to children and families. The reviewer does not seem to be aware of the reality of working with under-resourced communities Nonetheless we gave a flavour of the families' experiences and the other pressures they face. We would dispute and are annoyed by the reviewer's use of the word 'random' - this is life for people in these marginalised communities.
Comment 13: Line 363 – the many references to Whatsapp are redundant – it would be better to include a paragraph addressing communication strategies
Response 13: There are only two references to two different WhatsApp groups which are best understood in the specific context for which they were created rather than under a global heading as the reviewer suggest.
Comment 14; Lines 376-377 – I don’t understand how this relates to the larger article? Early intervention has been the focus but at this point income generation and vocational training is being discussed again without statistics addressing the need or linking it to early intervention.
Response 14: The first sentence in this section sets the scene with respect to poverty alleviation for families with a suitable citation provided. The reviewer seems to have forgotten that in the first paragraph of the paper, poverty is mentioned and in describing the country and community contexts in which the projects work, we highlighted the deprivation families experience. Hence income generation and vocational training is aimed at care-givers of the child.
Comment 15: Lines 390 – What was the transition plan or process>?
Response 15: We are unsure what the reviewer means by this comment. Our essential point was to show how the project linked with the educational psychologists and we described the school placements that resulted. It was beyond the scope of this article to detail the procedures used around school attendance.
Comment 16: Outputs and Outcomes – no methodology is disclosed for how these quotes were collected.
Response 16. We do describe how the information was collected in lines 409 to 411 and again in lines 413/414 and following. Figure 1 provides the detail. See response 12 for further clarification.
Comment 17: The discussion lacks a comparison to any other currently established programs in these communities or other similar areas of the world – the goals of the initiatives are broad.
Response 17: Not only in the discussion (with 9 citations) but throughout the article, we have cited pertinent literature relating to similar projects and policy statements. We are astounded and dismayed with the comment that the "goals of the initatives are broad" when we have given detailed description of the many specific goals that the projects addressed.
Comment 18: Conclusion – I do not think the procedures or steps taken to establish these initiatives were clearly stated therefore limiting their impact as a model for other parts of the world.
Response 18: We are astounded and dismayed that the reviewer could make such a comment given the 3,553 words we used to detail the work of the projects. S/he has left us speechless!